# Competing risk of mortality on loss to follow-up outcome among patients with HIV on ART: a retrospective cohort study from the Zimbabwe national ART programme

Zvifadzo Matsena Zingoni [1,2] Tobias Chirwa [1] Jim Todd [3] Eustasius Musenge [1]

¹Division of Epidemiology and Biostatistics, Faculty of Health Sciences, University of the Witwatersrand, Johannesburg, South Africa
²National Institute of Health Research, Ministry of Health and Child Care, Harare, Zimbabwe
³Department of Population Health, London School of Hygiene and Tropical Medicine, London, United Kingdom

**Correspondence to**
Zvifadzo Matsena Zingoni;
zmatsena28@gmail.com

## ABSTRACT

**Objective** To determine the loss to follow-up (LTFU) rates at different healthcare levels after antiretroviral therapy (ART) services decentralisation among ART patients who initiated ART between 2004 and 2017 using the competing risk model in addition to the Kaplan-Meier and Cox regressions analysis.

**Design** A retrospective cohort study.

**Setting** The study was done in Zimbabwe using a nationwide routinely collected HIV patient-level data from various health levels of care facilities compiled through the electronic patient management system (ePMS).

**Participants** We analysed 390 771 participants aged 15 years and above from 538 health facilities.

**Outcomes** The primary endpoint was LTFU defined as a failure of a patient to report for drug refill for at least 90 days from last appointment date or if the patient missed the next scheduled visit date and never showed up again. Mortality was considered a secondary outcome if a patient was reported to have died.

**Results** The total exposure time contributed was 1 544 468 person-years. LTFU rate was 5.75 (95% CI 5.71 to 5.78) per 100 person-years. Adjustment for the competing event independently increased LTFU rate ratio in provincial and referral (adjusted sub-HRs (AsHR) 1.22; 95% CI 1.18 to 1.26) and district and mission (AsHR 1.47; 95% CI 1.45 to 1.50) hospitals (reference: primary healthcare); in urban sites (AsHR 1.61; 95% CI 1.59 to 1.63) (reference: rural); and among adolescence and young adults (15–24 years) group (AsHR 1.19; 95% CI 1.16 to 1.21) (reference: 35–44 years). We also detected overwhelming association between LTFU and tuberculosis-infected patients (AsHR 1.53; 95% CI 1.45 to 1.62) (reference: no tuberculosis).

**Conclusions** We have observed considerable findings that 'leakages' (LTFU) within the ART treatment cascade persist even after the decentralisation of health services. Risk factors for LTFU reflect those found in sub-Saharan African studies. Interventions that retain patients in care by minimising any 'leakages' along the treatment cascade are essential in attaining the 90–90–90 UNAIDS fast-track targets.

## Strengths and limitations of this study

► The analysis is based on a huge national routinely collected individual-level dataset that has a long follow-up period making our findings generalisable to the patients with HIV on antiretroviral therapy in Zimbabwe.

► This study fitted a competing risk model based on the cumulative incidence function to explore the association between selected covariates and the absolute risk, which is an essential modelling approach for medical decision making so that appropriate subpopulation interventions can be implemented.

► Variable selection of the numerous independent factors was performed using the regularisation (shrinkage) lasso regression technique alongside the stepwise section methods.

► Important factors known to influence LTFU like CD4 cell counts and viral load were not accounted for in the model due to the high percentage of missingness, and the independent factors adjusted for were not time dependent.

► This study has a possibility of participant selection bias since only participants enrolled in health facilities linked to the electronic patients' management database were used in the analysis.

## INTRODUCTION

Globally, remarkable progress has been made on the treatment of HIV-infected people using antiretroviral therapy (ART), particularly in sub-Saharan Africa (SSA).[1] In Zimbabwe, the ART programme is a comprehensive care and support package aiming to slow disease progression,[2] improve survival and quality of life[3] and minimise HIV transmission by reducing the viral load. Therefore, it is imperative to monitor individuals on ART to ensure treatment efficacy, support adherence and identify treatment failure and resistance.

At the national level, aggregated health facility data are usually used for monitoring the ART programme. However, there is an inherent difficulty in using aggregated data to make statistical inferences about individual-level outcomes. This promoted the increasing use of patient-level data worldwide to assess the impact of the ART programme.[4–8] With efforts to achieve zero HIV incidences by 2030, the 90–90–90 UNAIDS targets were launched, whereby 90% of the people living with HIV (PLWHIV) should know their status, 90% of those diagnosed should be initiated on ART and 90% of those on ART should achieve viral suppression.[9] In Zimbabwe, strides have been made towards achieving these targets; however, limitations exist. For instance, in Zimbabwe, only 73% of PLWHIV who are on ART have achieved viral suppression.[10] To improve the proportion of those who achieve viral suppression, it is crucial to come up with broad efforts that minimise long-term ART attrition due to loss to follow-up (LTFU), mortality, drop-outs and withdrawals.

In Zimbabwe, the health service delivery can be categorised into four levels of care, namely the primary, secondary, tertiary and quaternary.[11] The primary care consists of clinics and facilities offering basic preventive and curative services. Majority of these primary healthcare (PHC) facilities are in the rural areas, and health problems beyond the scope of these facilities are referred to the secondary healthcare centres, which are mainly the district and mission hospitals. In the secondary care category, private and company health facilities may fit in if they can handle emergency services as the district hospitals. The tertiary care consists of the provincial hospitals that received referral patients from the district hospitals.[12] The quaternary care is the last referral care level that received patients from provincial hospitals. These facilities are mainly in the two largest cities in Zimbabwe.[12] The quaternary healthcare offers much more advanced healthcare; hence, there is modern equipment, medical specialists and pharmaceutical, which could be missing in the lower levels of care.[11]

The ART services were initially centralised at the higher levels of care—quaternary and tertiary healthcare facilities—normally situated in urban areas. This means rural residents and poorer people were less likely to access HIV testing, care and treatment due to transport cost. In 2010/2011, one of the strategies to mitigate the transport barrier was the decentralisation of ART services to lower levels of care.[13] These devolved care models were implemented in Zimbabwe to improve patients' access to HIV care, ART adherence and retention in care of PLWHIV. The impact of HIV care and ART decentralisation has been well documented in the SSA region.[14] Understanding ART retention patterns after the implementation of decentralised care is vital to guide the ART programme and policy as well as reflecting on the gaps that may contribute to the poor viral suppression. LTFU is one of the main threats to patient retention and may persist even after the decentralisation of ART services.

This may be due to social (stigmatisation, social deprivation and health literacy issues) and individual factors (perception and misconceptions over ART benefits) that still hinder retention in care.[15]

This study aims to determine LTFU rates at different healthcare levels after ART services decentralisation among ART patients who initiated ART between 2004 and 2017. The study period covers the pre-ART and post-ART decentralisation period to get an overall picture of the LTFU patterns over time. The LTFU rates were compared between the different healthcare facility types (PHC, district/mission and provincial/referral hospitals) in Zimbabwe using routinely collected patients-level data from the Zimbabwe national ART database. We further adjusted for the individual-level demographic and clinical characteristics as predisposing or independent factors for LTFU using a statistical model that adjusted for the competing risk (CR) of mortality on the LTFU outcome.

## METHODS

### Study population

This study was conducted in Zimbabwe, a landlocked country in SSA with roughly 1.4 million PLWHIV. The HIV prevalence dropped from 25% in 2004 to 13.7% in 2017 among adults.[10 16] Individual patients aged 15 years and above, who initiated ART between 1 January 2004 and 31 December 2017, were eligible for this analysis. A full description of how the final sample size was achieved is illustrated in figure 1.

### Study design and data sources

This study was a secondary data analysis from a retrospective cohort study of routinely collected patient data from the Zimbabwe national ART programme. The data came from the national ART programme, which started in 2004. The ART sites expanded from 5 health facilities in 2004 to 1566 by 2017.[10 17] Treatment guidelines followed WHO recommendations; hence, the ART eligibility criteria and treatment regimens changed over time. The initial regimen for an ART-naïve patient is the first-line ART, a combination of two nucleoside reverse transcriptase inhibitors (NRTIs), the first drug that can be zidovudine, tenofovir (TDF) (the most common) or stavudine (D4T) and the second drug is mainly lamivudine (3TC), and one non-NRTI that can either be nevirapine (NVP) or efavirenz (EFV). Majority of the participants were receiving three-drug combination therapy of TDF+3TC+EFV therapy as the first-line treatment. Once the patient experienced treatment failure or intolerance, switch to second-line ART (a three-drug combination of protease inhibitor and two NRTIs) can be made.[18]

### Monitoring and evaluation system

The monitoring and evaluation system of the ART programme in Zimbabwe has been described elsewhere.[19] Briefly, since the inception of ART, all patient monitoring

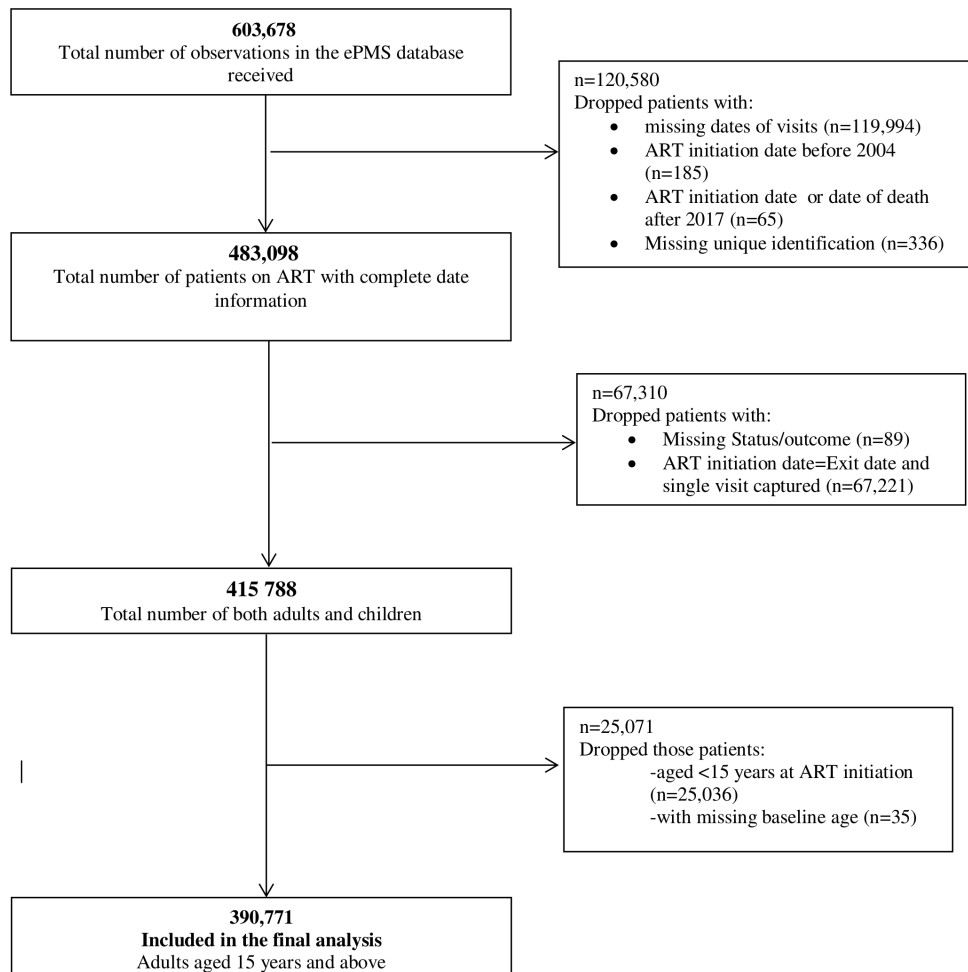

**Figure 1** Flow chart of the inclusion criteria for analyses of adult ART patients enrolled in Zimbabwe ART Programme followed up between 2004 and 2017. ART, antiretroviral therapy; ePMS, electronic patient management system.

records had been collected manually using a paper-based system and kept at the health facility where the patient is registered. Since 2012, the paper-based system was no longer functioning properly due to high volumes of ART patients.[10] This upsurge in patients' volumes began to affect the accurate ART monitoring and reporting process as the paper-based system could not cope. Moreover, the paper-based system translated into huge workloads and became strenuous to the already overburdened health workers. As a result, the electronic patient management system (ePMS) was launched in 2012. The ePMS increased efficiency and effective management of ART patients through improving follow-ups to subsequently increase ART adherence.[19] The ePMS provided a platform to compile quality patient-level data to accurately make forecast and evaluate the ART intervention, enabling a more effective programme.[10]

The ePMS was rolled out in systematic phased approach with priority given to those HIV sites with high volumes of patients with HIV at the same time maintaining a good representation of the four healthcare levels in the country.[19] From the ePMS Strategic Plan document, the first phase target was 83 sites in 2013 comprising of central, provincial and district hospitals that covered approximately 61% of ART patients nationwide. Additional 267 city polyclinics, rural/large mission hospitals sites were included to make a total of 350 sites by the end of 2014.[19] The last phase included 184 static ART follow-up sites to bring the total to 534 sites by the end of 2017. By the end of 2018, there were 622 sites with a functional ePMS with the inclusion of private hospitals, which meant that approximately 95% of all patients receiving ART were linked to the ePMS database.[10]

The final database of the ePMS has patients' information from all levels of care (quaternary, tertiary, secondary and primary healthcare levels). It, therefore, reflects the overall ART management services in the country, although the database includes all HIV-related data. At enrolment, the patient's demographic data are captured. At each subsequent clinic visit, information on ART regimen type, clinical information, laboratory investigations and physical examination is captured. Outcomes are assessed and captured as death, LTFU, transfer-out to another health facility or alive and active on treatment. Already existing patient's information prior to 2012 from the paper-based system was retrospectively transcribed into the electronic database at each clinic, and all subsequent clinic visits were recorded prospectively with a real-time entry of patient records.

The introduction of the ePMS for monitoring patients with HIV in 2012 has provided a platform for huge data repository in the country that may be used to answer some of the public health questions on the HIV aspects using statistical models. Moreover, the ePMS platform provides readily available data archiving for robust statistical analysis approaches to guide evidence-based decision making and programme interventions planning at patients level while eliminating the aggregated data. Therefore, this rich, huge patient-level database was of preference and suitable to answer our study objective.

## Study endpoints

The primary endpoint was LTFU defined as a failure of a patient to report for drug replenishment within 90 days from last appointment date or if the patient missed the next scheduled visit date and never showed up again. Patients, who returned after having been LTFU, were classified as alive and active on treatment. We also consider the mortality outcome as a competing event of LTFU if a patient was reported to have died.

## Data analysis methods

Descriptive summaries were reported as frequencies and proportions stratified by health facility type. The pattern of LTFU was explored by calendar year. Incidence rates were calculated from the ART initiation date to outcome occurrence date on a quarterly time scale. Kaplan-Meier non-parametric curves (equations 1–4 in online supplemental appendix) were used to describe the probability of LTFU over time stratified by selected covariates. The log-rank test was used to assess the equality of these patient retention probabilities. Estimated hazard rates (HRs) (equations 5–9 in online supplemental appendix) were reported as per 100 person-years.

Important variable selection was done using the lasso regression regularisation technique. Variable selection sensitivity analysis was done using the subset selection techniques (best subset selection, stepwise forward variable selection and stepwise backward variable selection methods). All these methods yield the same independent factors considered in the seven different multiple adjusted time-to-event models fitted (figure 2). The baseline predictors used for adjustment were all categorical, namely: health facility type, degree of urbanisation of the health facility (rural/urban), age groups (15–24, 25–34, 35–44 and 45 and above), tuberculosis status (positive/negative), body functional status—which is the patient's ability to perform normal daily activities expected for the well-being of an individual—(working body, ambulatory body and bed-ridden body), WHO clinical stages (stage 1, II, III and IV), ART enrolment calendar period (2004–2007; 2008–2011; 2012–2014; and 2015–2017) and regimen type (D4T, AZT or TDF based combination

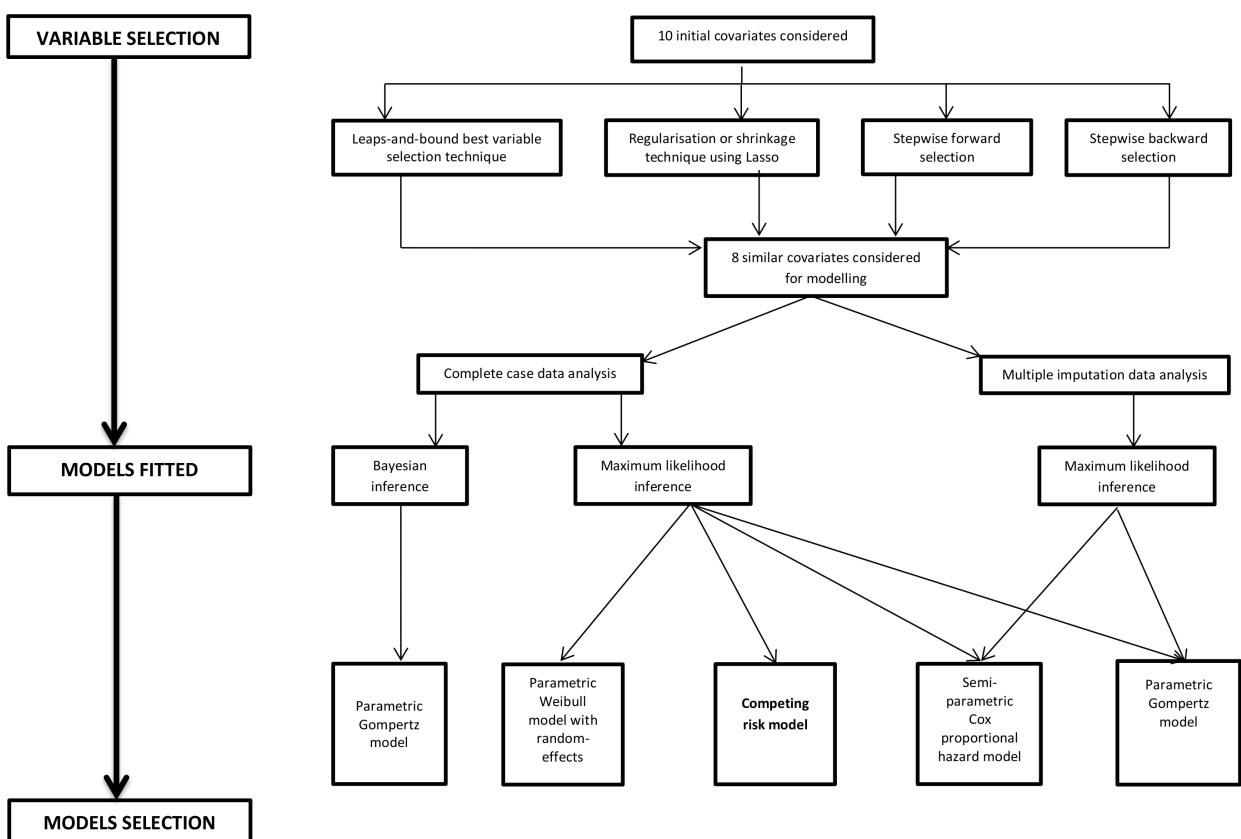

**Figure 2** Schematic illustration of the variable of importance selection steps and subsequent models fitted for sensitivity analysis of the results.

therapy). The stratification of the health facility type into primary healthcare, district/mission hospitals and provincial/referral hospitals was based on the level of care classification of the corresponding health facility an individual was enrolled in, and the grouping of the health facilities was based on the patients' care similarities. For those individuals who had moved from one health facility to the other during the study period covered, the last recorded health facility site was considered for this analysis. Therefore, the health level of care classification was mutually exclusive as one individual was assigned to the last captured health facility.

The standard Cox regression model defined in equations 10–12 in the online supplemental appendix is normally used for a time-to-event analysis; however, this method may give biased estimates in the presence of competing events. We fitted the CR model of mortality on LTFU through the cumulative incidence function defined in equations 13–18 in online supplemental appendix.[20] CR analysis assumes that patients are exposed to at least two risks (mortality and LTFU). One of the events is of interest (LTFU), while the other is a competing event (mortality) that inhibits the outcome of interest from occurring. The competing events may not be independent, and the baseline hazard may differ between them. The covariates used were time independent; however, we adjusted for time to take into account the period the country experienced an economic turmoil around 2008, which resulted in a mass withdrawal of health personnel in the public sectors and an increased scarcity of vital medical supplies in health system including HIV medication.[21 22] The adjusted sub-HRs (AsHR) for covariates on LTFU were reported with 95% CIs. We censored on 31 December 2017 for those who had not experienced the event of interest and at the last observed date for those who had transferred to other health facilities.

We performed a sensitivity analysis of our key findings through multiple imputations of the missing explanatory covariates using Rubin's method, which assumes that the data are missing at random.[23] We compared imputed data and the complete case data estimates based on the maximum likelihood estimation (MLE) perspective for both the Cox proportional hazard and Gombertz parametric models. Within the Gombertz parametric models, estimates from the Bayesian estimation (BE) and MLE were compared.[24] We assumed non-informative normal priors for all unknown parameters to be estimated with a mean of zero and variance of 100 or precision of 0.001 for the parametric Bayesian model. We fitted a parametric Weibull model with random effects to account for any individual heterogeneity.[25] Model comparison between BE and MLE was done using the Akaike's information criterion (AIC), Bayesian information criterion (BIC) and deviance information criterion. In contrast, model comparison between complete case analysis and imputed data analysis was done using the precision of the CI for the estimates. All data management and analysis were performed using Stata V.15.1.[26]

## Patient and public involvement

This study acknowledges the importance of patients and public involvement in research studies; however, this research was done without patient involvement in all research stages since it was secondary data analysis.

## Ethical approval

Since this study was secondary data analysis, we did not have direct contact with the participants. We received only anonymised data for analyses.

## RESULTS

### Descriptive analysis

At the end of study, 268 896 (68.81%) patients were alive, 18 328 (4.7%) had died, 88 744 (22.71%) were LTFU and 14 803 (3.79%) had transferred out from the initial health facility another health facility. Over 1 544 468 person-years of ART, mortality and LTFU rates were 1.19 (95% CI 1.17 to 1.20) and 5.75 (95% CI 5.71 to 5.78) per 100 person-years, respectively. The quarterly trend analysis graph of LTFU over time showed a general increase in LTFU over time with significant peaks in 2012, 2014 and 2016 (figure 3).

The descriptive characteristics of 390 771 patients from 538 health facilities are presented in table 1. The study participants were enrolled at PHC (45.1%), districts/ mission (49.2%) or provincial/referral (5.7%) hospitals. Most patients were enrolled in rural health facilities (66.52%). At ART initiation, the majority of the patients were female 255 844 (65.47%), the mean (SD) age was 37.5 (10.2) years and only 6.44% (n=25, 143) patients had either an ambulatory (n=24,332; 6.23%) or a bed-ridden (n=811; 0.21%) body functional status.

Most patients were married (59.5%) with secondary education (31.8%), while 54.2% were classified in WHO stage III or IV and 34.18% were in the 35–44 years age group. The majority of the participants were on first-line ART treatment (97.7%), and only a few had confirmed tuberculosis infection (1.2%). There were missing data on some covariates, including baseline CD4 cell counts recorded for only 35.3% of the study participants.

The time-to-event rates for LTFU stratified by baseline characteristics are shown in table 2. LTFU rate per 100 person-years was 5.92 for patients enrolled at district and mission hospitals, 6.60 among patients from rural health facilities and 17.09 among patients diagnosed with tuberculosis.

The cumulative incidence probability (CIP) curve for death was lower than the CIP for LTFU as the outcome (figure 4A). The LTFU CIP curve increased in the first 3 years on ART and became steady after that. We observed overwhelmingly significant variation between LTFU and selected covariates, log-rank p value <0.001. Higher risk of LTFU was observed among patients aged 15–24 years (figure 4B), females (figure 4C), patients enrolled at urban health facilities (figure 4D), tuberculosis-infected

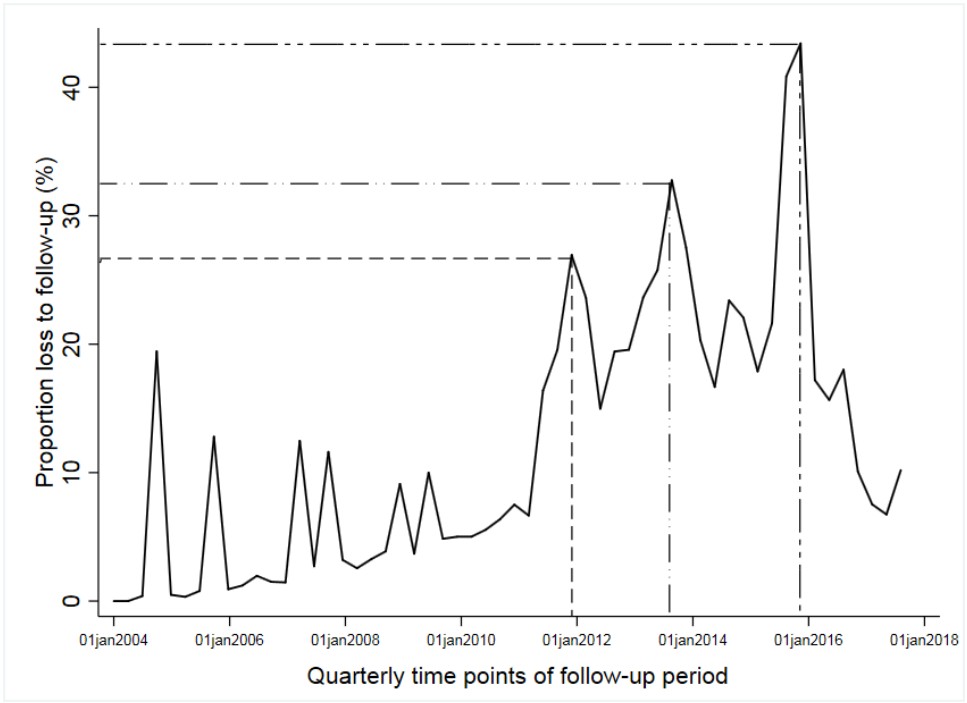

**Figure 3** Trend analysis of loss to follow-up outcome among patients receiving ART in Zimbabwe national ART programme on a quarterly time scale, 2004–2017.

patients (figure 4E) and WHO stage I/II patients (figure 4F).

### LTFU outcome

To determine the possible independent risk factors associated with LTFU, seven models were fitted, and the results are shown in table 3. An additional pictorial view of the estimates is provided in the online supplemental appendix figure A1 to compare the different model estimates. The data had approximately 10% of missing information in the explanatory covariates. There was no difference in the model estimates between imputed data and complete case data (model 1 vs model 2 and model 4 vs model 5). Comparing the MLE (model 3) and the BE (model 4) models, the estimates were almost similar for most regression parameters, and inclusion of the individual random effects did not improve the model estimates (model 4 vs model 6). Lastly, adjusting for the competing effects of mortality on LTFU showed some significant improvement to the model based on the reduced AIC and BIC values compared with the Cox proportional hazard model. Hence, the CR model was considered a better fit compared with other models, and the interpretation of the results will be based on this model.

The multivariable CR model revealed that patient factors predicting LTFU included: being enrolled at a provincial/referral (AsHR 1.22; 95% CI 1.18 to 1.26) or district/mission (AsHR 1.47; 95% CI 1.45 to 1.50) hospitals (reference: PHC), urban sites (AsHR 1.61; 95% CI 1.59 to 1.63) (reference: rural) and 15–24 years group (AsHR 1.19; 95% CI 1.16 to 1.21) (reference: 35–44 years). We also detected an overwhelming association between LTFU and tuberculosis-infected patients (AsHR

1.53; 95% CI 1.45 to 1.62) (reference: no tuberculosis) and a protective effect of LTFU for patients classified in WHO stage III or IV (AsHR 0.74; 95% CI 0.73 to 0.75) (reference: WHO I/II). Patients who were taking stavudine combination therapy as a first-line ART treatment had an increased risk of becoming LTFU relative to patients taking TDF+3TC+EFV first-line combination therapy, that is, D4T(30)+3TC+NVP (AsHR 5.47; 95% CI 5.52 to 5.75) and D4T(30)+3TC+EFV (AsHR 7.98; 95% CI 6.41 to 9.94). We also observed that patients who initiated ART between 2015 and 2017 had a pronounced risk of becoming LTFU compared with patients who enrolled between 2012 and 2014 (AsHR 6.02; 95% CI 5.91 to 6.12).

### DISCUSSION

This study aims to determine the LTFU rates at different healthcare levels after ART services decentralisation among ART patients who initiated ART between 2004 and 2017 and determine other potential independent factors using the CR model approach. Our study followed similar studies adjusting for the competing effect of mortality on LTFU, recognising that those who die are no longer at risk of LTFU.[27 28] We found the LTFU rate of 5.7 per 100 person-years, which was lower than that reported in Ethiopia of 8.2 per 100 person-years.[29] The lower LTFU rates in our study put forward the positive impact of the decentralised models implemented in Zimbabwe to increase patients' retention in HIV care. However, there is still a considerable magnitude of LTFU outcome in ART programmes contributing to the 'leakages' or attrition of patients within the continuum of care cascade in

**Table 1** Patients' sociodemographic and clinical characteristics at ART initiation in Zimbabwe national ART programme, 2004–2017

| Characteristics | Categories | Primary healthcare (PHC) n=176 253 (45.1%) (n (%)) | District/mission hospitals n=192 238 (49.2%) (n (%)) | Provincial/referral hospitals n=22 280 (5.7%) (n (%)) | Total n=390 771 (100%) (n (%)) |
|---|---|---|---|---|---|
| Patient demographics | | | | | |
| Age (years) at ART initiation | 15–24 | 20 314 (11.53) | 21 057 (10.95) | 2601 (11.67) | 43 972 (11.25) |
| | 25–34 | 54 416 (30.87) | 58 874 (30.63) | 6979 (31.32) | 120 269 (30.78) |
| | 35–44 | 59 284 (33.64) | 66 418 (34.55) | 7875 (35.35) | 133 577 (34.18) |
| | 45+ | 42 239 (23.96) | 45 889 (23.87) | 4825 (21.66) | 92 953 (23.79) |
| Degree of urbanisation | Rural | 113 738 (66.92) | 137 380 (73.57) | 432 (2.01) | 251 550 (66.52) |
| | Urban | 56 223 (33.08) | 49 348 (26.43) | 21 055 (97.99) | 126 626 (33.48) |
| Sex | Female | 117 567 (66.70) | 124 141 (64.58) | 14 136 (63.45) | 255 844 (65.47) |
| | Male | 58 686 (33.30) | 68 097 (35.42) | 8144 (36.55) | 134 927 (34.53) |
| Education status at ART initiation | None | 3854 (2.19) | 4136 (2.15) | 227 (1.02) | 8217 (2.10) |
| | Primary | 33 735 (19.14) | 34 062 (17.72) | 2248 (10.09) | 70 045 (17.92) |
| | Secondary | 59 181 (33.58) | 58 349 (30.35) | 6593 (29.59) | 124 123 (31.76) |
| | Tertiary | 2921 (1.66) | 4423 (2.30%) | 1177 (5.28) | 8521 (2.18) |
| | Missing | 76 562 (43.44) | 91 268 (47.48) | 12 035 (54.02) | 179 865 (46.03) |
| Marital status at ART initiation | Single | 24 438 (13.87) | 24 489 (12.74) | 3530 (15.84) | 52 457 (13.42) |
| | Married | 105 504 (59.86) | 113 953 (59.28) | 12 935 (58.06) | 232 392 (59.47) |
| | Widowed | 25 798 (14.64) | 30 994 (16.12) | 3523 (15.81) | 3523 (15.81) |
| | Divorced | 13 429 (7.62) | 14 732 (7.66) | 1677 (7.53) | 29 838 (7.64) |
| | Missing | 7084 (4.02) | 8070 (4.20) | 615 (2.76) | 15 769 (4.04) |
| Calendar time of starting ART | 2004–2007 | 6635 (3.76) | 8942 (4.65) | 2962 (13.29) | 18 539 (4.74) |
| | 2008–2011 | 38 990 (22.12) | 61 321 (31.85) | 7899 (35.45) | 108 120 (27.67) |
| | 2012–2014 | 67 759 (38.44) | 90 582 (47.12) | 9472 (42.51) | 167 813 (42.94) |
| | 2015–2017 | 62 869 (35.67) | 31 483 (16.38) | 1947 (8.74) | 96 299 (24.64) |
| Clinical factors | | | | | |
| WHO staging at ART initiation | I/II | 82 203 (46.64) | 69 629 (36.22) | 7690 (34.52) | 159 519 (40.82) |
| | III/IV | 79 279 (44.98) | 118 113 (61.44) | 14 425 (64.74) | 211 817 (54.20) |
| | Missing | 14 771 (8.38) | 4499 (2.34) | 165 (0.74) | 19 435 (4.97) |
| Baseline CD4 (cell/µL) at ART initiation | Above 500 | 4783 (2.71) | 4950 (2.57) | 601 (2.70) | 10 334 (2.64) |
| | 351–500 | 8227 (4.67) | 8790 (4.57) | 940 (4.22) | 17 957 (4.60) |
| | 200–350 | 17 908 (10.16) | 24 895 (12.95) | 2866 (12.86) | 45 669 (11.69) |
| | Below 200 | 23 796 (13.50) | 35 020 (18.22) | 4982 (22.36) | 63 798 (16.33) |
| | Missing | 121 539 (68.96) | 118 583 (61.69) | 12 891 (57.86) | 253 013 (64.75) |
| Treatment regimen at the time of data extraction | First line | 170 662 (98.29) | 184 321 (97.37) | 20 724 (94.32) | 375 704 (97.65) |
| | D4T(30)+3TC+NVP | 625 (0.35) | 2942 (1.53) | 629 (2.82) | 4196 (1.07) |
| | D4T(30)+3TC+EFV | 42 (0.02) | 155 (0.08) | 20 (0.09) | 217 (0.06) |
| | AZT+3TC+NVP | 1349 (0.77) | 1933 (1.01) | 349 (1.57) | 3631 (0.93) |
| | AZT+3TC+EFV | 212 (0.12) | 425 (0.22) | 56 (0.25) | 693 (0.18) |
| | TDF+3TC+NVP | 6541 (3.71) | 16 334 (8.50) | 2923 (13.12) | 25 798 (6.60) |
| | TDF+3TC+EFV | 161 288 (91.51) | 162 006 (84.27) | 16 628 (4.63) | 339 922 (86.99) |
| | Other first lines | 3236 (1.84) | 3474 (1.81) | 428 (1.92) | 7138 (1.83) |
| | Second line | 2960 (1.68) | 4969 (2.58) | 1247 (5.60) | 9176 (2.35) |
| Tuberculosis status at ART initiation | Negative | 156 072 (88.55) | 170 869 (88.88) | 20 218 (90.75) | 347 159 (88.84) |
| | Positive | 2092 (1.19) | 2423 (1.26) | 301 (1.35) | 4816 (1.23) |
| | Not assessed | 15 743 (8 93) | 16 387 (8.52) | 1347 (6.05) | 33 477 (8.57) |
| | Missing | 2346 (1.33) | 2559 (1.33) | 414 (1.86) | 5319 (1.36) |

**Table 1** Continued

| Characteristics | Categories | Primary healthcare (PHC) n=176 253 (45.1%) (n (%)) | District/mission hospitals n=192 238 (49.2%) (n (%)) | Provincial/referral hospitals n=22 280 (5.7%) (n (%)) | Total n=390 771 (100%) (n (%)) |
|---|---|---|---|---|---|
| Body functional status at ART initiation | Working | 162 710 (92.32) | 180 417 (93.85) | 20 024 (89.87) | 363 151 (92.93) |
| | Ambulatory | 11 789 (6.69) | 10 454 (5.44) | 2089 (9.38) | 24 332 (6.23) |
| | Bed ridden | 331 (0.19) | 452 (0.24) | 28 (0.13) | 811 (0.21) |
| | Missing | 1423 (0.81) | 915 (0.48) | 139 (0.62) | 2477 (0.63) |

Missing category refers to information that is unavailable in the dataset for that particular variable.
ART, antiretroviral therapy; AZT, zidovudine; D4T, stavudine; EFV, efavirenz; NVP, nevirapine; 3TC, lamivudine; TDF, tenofovir.

Zimbabwe. This requires continual upscaling of interventions that minimise the LTFU rates and retain patients in care since patients who are LTFU are associated with increased treatment interruptions that can trigger ART treatment defaulting, treatment non-adherence, treatment failure or treatment resistance.[30 31]

Similar to other studies, LTFU independently increased at higher levels of care (provincial and referral (AsHR 1.22; 95% CI 1.18 to 1.26) and district and mission (AsHR 1.47; 95% CI 1.45 to 1.50) hospitals) relative to PHC in this study.[5] This finding supports the fact that decentralisation is associated with better patients' retention in care as there are high risks of LTFU rate in higher levels of care compared with the PHC facilities. This finding supports the assumption that if patients seek care from health facilities closer to their homes with cheaper transport cost and less congested hospital environments, there is improved patient retention as compared with seeking care to faraway health facilities.[32] Moreover, the high rates of LTFU in secondary and tertiary/quaternary levels of care could be explained by 'silent transfers' whereby patients self-transfer themselves to these decentralised PHC facilities based on individual preferences of where to seek care, and the system fails to track it. These 'silent transfers' could be a good indication of the upsurge of the decentralisation of services.[32] However, such 'silent transfers' may result in inflated LTFU rates in higher levels of care as most these patients would be classified as LTFU when in fact they might have self-transferred and the patient tracking system is not efficient enough to pick such scenarios due to significantly high patients' volumes persisting at these health facilities.[5] Similarly, the quarterly trend analysis of LTFU rates showed an increase over time, peaking in the period when decentralised models were implemented. The same issue of 'silent transfers' of patients to nearer health facilities can also explain this trend pattern finding.

On the contrary, not everyone would prefer to seek care from these decentralised health facilities. Some may intentionally avoid these nearby health facilities to conceal their HIV status from the community. These individuals remain enrolled in faraway health facilities and most probably the higher levels of care. These individuals may contribute to the observed high rates of LTFU in this cohort as they may still face transport cost challenges to replenish their ART supplies.[33] Moreover, tertiary and quaternary levels of care may receive much sicker patients with a high risk of death, hoping to get better treatment through the healthcare referral system.[34] This means some of these much ill patients are more likely to become LTFU eventually as severe immune deterioration hinders one to keep up to date with their treatment replenishments. Earlier studies have also reported that much sicker patients with severe immune deterioration are more likely to become LTFU compared with those with no significant immunodeficiency.[35] Other studies have also shown that a significant portion of these severely ill patients who eventually become LTFU from higher levels of care health facilities, if tracked, could be found dead.[36] Our findings put forward that though the decentralisation uptake might have been taken up by the majority of the ART patients and improved patient's retention in the ART programme in overall; those groups of individuals who still seek healthcare from the higher levels of care are still more at risk of becoming LTFU; hence, interventions that may retain patients in the higher levels of care facilities are a priority.

Patients enrolled at urban health facilities had increased risk to become LTFU compared with those at rural sites. This finding contradicts a study done in Ethiopia whereby rural residents had an increased risk of LTFU due to availability of transport costs, long distances to ART health facilities, social stigma as well as low levels of HIV knowledge in these communities.[15] However, our study finding can be explained by the decentralisation of health facilities to walkable distances in favour of rural residents.[4 37] Mobility of residents in urban areas is disproportionate relative to rural residents; hence, most of these LTFU in urban areas could be part of the silent transfers to other nearer ART health facilities in their relocated areas.

Another interesting finding was high LTFU among the 15–24 years group. The adolescent's group is known to be associated with additional intricacy in their HIV management due to different types of chronic comorbidities, and this whole complexity may impact on their ART outcomes including LTFU.[38 39] The high risk of becoming LTFU among adolescents can be attributed to structural deprivation due to medical and psychological reasons that contribute to poor adherence in this subpopulation group as a result of becoming LTFU.[40] The most common

**Table 2** Rates per 100 person-years of LTFU and log-rank tests for ART patients' baseline characteristics in Zimbabwe national ART programme, 2004–2017

| Characteristics | Category | LTFU cases | Person-years | Rate per 100 person-years | 95% CI of rate | Log-rank p value |
|---|---|---|---|---|---|---|
| Health facility level | Primary healthcare | 34 958 | 6200 | 5.6502 | 5.5913 to 5.7097 | <0.001* |
| | District/mission hospitals | 48 188 | 8100 | 5.9169 | 5.8643 to 5.9700 | |
| | Provincial/referral hospitals | 5598 | 1100 | 5.0273 | 4.8973 to 5.1608 | |
| Age (years) at ART initiation | 15–24 | 32 790 | 4500 | 7.3252 | 7.2463 to 7.4049 | <0.001* |
| | 25–34 | 12 617 | 1500 | 8.4301 | 8.2843 to 8.5785 | |
| | 35–44 | 27 504 | 5500 | 4.9671 | 4.9087 to 5.0261 | |
| | 45+ | 15 833 | 3900 | 4.0242 | 3.9620 to 4.0874 | |
| Degree of urbanisation | Rural | 52 629 | 10 000 | 5.2756 | 5.2307 to 5.3209 | <0.001* |
| | Urban | 36 115 | 5500 | 6.6039 | 6.5362 to 6.6724 | |
| Sex | Females | 59 695 | 10 000 | 5.8703 | 5.8234 to 5.9175 | <0.001* |
| | Male | 29 049 | 5300 | 5.5063 | 5.4433 to 5.5700 | |
| Education status at ART initiation | None | 1470 | 354 | 4.1582 | 3.9509 to 4.3762 | <0.001* |
| | Primary | 14 786 | 2800 | 5.3558 | 5.2702 to 5.4429 | |
| | Secondary | 32 629 | 4700 | 6.9961 | 6.9206 to 7.0725 | |
| | Tertiary | 1803 | 409 | 4.4097 | 4.2108 to 4.6180 | |
| | Missing | 38 056 | 7300 | 5.2435 | 5.1911 to 5.2965 | |
| Marital status at ART initiation | Single | 13 078 | 1900 | 6.9036 | 6.7863 to 7.0230 | <0.001* |
| | Married | 7548 | 1100 | 6.6112 | 6.4637 to 6.7620 | |
| | Widowed | 9375 | 2900 | 3.1872 | 3.1233 to 3.2524 | |
| | Divorced | 55 985 | 8800 | 6.3313 | 6.2790 to 6.3839 | |
| | Missing | 2758 | 625 | 4.416 | 4.2543 to 4.5840 | |
| Calendar time of starting ART | 2004–2007 | 790 | 1900 | 0.4255 | 0.3968 to 0.4562 | <0.001* |
| | 2008–2011 | 4572 | 6900 | 0.6593 | 0.6405 to 0.6787 | |
| | 2012–2014 | 43 565 | 5300 | 8.1802 | 8.1038 to 8.2574 | |
| | 2015–2017 | 39 817 | 1300 | 29.9903 | 29.6972 to 30.2863 | |
| WHO staging at ART initiation | I/II | 47 410 | 5100 | 9.208 | 9.1255 to 9.2912 | <0.001* |
| | III/IV | 38 919 | 9500 | 4.0957 | 4.0552 to 4.1366 | |
| | Missing | 2415 | 794 | 3.0432 | 2.9242 to 3.1671 | |
| Baseline CD4 (cell/uL) at ART initiation | Above 500 | 4341 | 257 | 16.919 | 16.4231 to 17.4299 | <0.001* |
| | 351–500 | 7614 | 455 | 16.7362 | 16.3645 to 17.1164 | |
| | 200–350 | 11 556 | 1700 | 6.7428 | 6.6210 to 6.8669 | |
| | Below 200 | 15 271 | 2700 | 5.7338 | 5.6436 to 5.8254 | |
| | Missing | 49 962 | 10 000 | 4.8245 | 4.7823 to 4.8669 | |
| Treatment regimen at the time of data extraction | First line | 88 050 | 15 000 | 5.9066 | 5.8677 to 5.9457 | <0.001* |
| | D4T(30)+3TC+NVP | 2685 | 168 | 15.9847 | 15.3914 to 16.6009 | |
| | D4T(30)+3TC+EFV | 141 | 10 | 14.7353 | 12.4932 to 17.3797 | |
| | AZT+3TC+NVP | 448 | 184 | 2.4374 | 2.2218 to 2.6738 | |
| | AZT+3TC+EFV | 158 | 30 | 5.3159 | 4.5484 to 6.2129 | |
| | TDF+3TC+NVP | 9652 | 1100 | 9.1737 | 8.9925 to 9.3586 | |
| | TDF+3TC+EFV | 71 378 | 13 000 | 5.371 | 5.3318 to 5.4106 | |
| | Other first lines | 3588 | 175 | 20.559 | 19.8972 to 21.2429 | |
| | Second line | 694 | 538 | 1.29 | 1.1985 to 1.3908 | |

Continued

**Table 2** Continued

| Characteristics | Category | LTFU cases | Person-years | Rate per 100 person-years | 95% CI of rate | Log-rank p value |
|---|---|---|---|---|---|---|
| Tuberculosis status at ART initiation | Negative | 81 119 | 14 000 | 5.9418 | 5.9010 to 5.9828 | <0.001* |
| | Positive | 2235 | 131 | 17.0906 | 16.3966 to 17.8141 | |
| | Not assessed | 4188 | 1400 | 2.9328 | 2.8453 to 3.0230 | |
| | Missing | 1202 | 234 | 5.1455 | 4.8627 to 5.4447 | |
| Body functional status at ART initiation | Working | 81 122 | 14 000 | 5.6445 | 5.6057 to 5.6834 | <0.001* |
| | Ambulatory | 7059 | 930 | 7.5897 | 7.4147 to 7.7688 | |
| | Bed ridden | 215 | 26 | 8.4201 | 7.3666 to 9.6243 | |
| | Missing | 348 | 117 | 2.9723 | 2.6758 to 3.3015 | |

*Significant at 5%. Missing category refers to information that is unavailable in the dataset for that particular variable.
ART, antiretroviral therapy; AZT, zidovudine; D4T, stavudine; EFV, efavirenz; LTFU, loss to follow-up; NVP, nevirapine; 3TC, lamivudine; TDF, tenofovir.

medical barrier that has been found to affect ART adherence include treatment side effects, multiple doses and health system dissatisfaction. Research has found that ART has improved perinatal HIV infection into a controllable disease through optimal ART adherence. This opens up an additional challenge among adolescents who were prenatally or perinatally infected in the transition stage to adult care to experience ART adherence lapses and develop worse clinical outcomes as they become LTFU.[40] The transition from the paediatric care to adult care would mean changing care providers, absence of youth-friendly facilities, tight and hectic schedules, new responsibilities and deprivation of caregiver support for some of these adolescents. Some of the social deprivation issues

among adolescents include inadequate financial support to cover transport costs to health facilities for treatment refill since this group heavily depends on their caregivers.[41] Some of the social deprivations faced by adolescents include lack of family support as some adolescents come from disrupted family structures with depraved caregiver–child relationships, fear-perceived HIV-related stigma and discrimination to disclose their status.[15 41 42]

Therefore, this study recommends that policymakers should strengthen the social protection support to address the complex vulnerability, disadvantages and risks faced by HIV-positive adolescents in order to foster resilience among the HIV-positive adolescents population. The social protection support can interrupt some of the

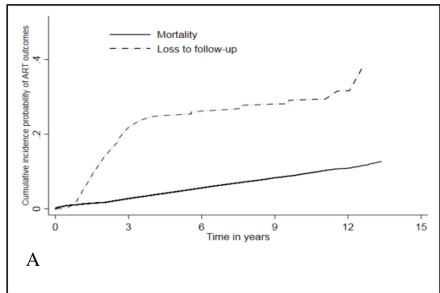

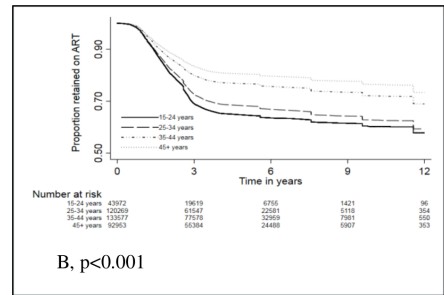

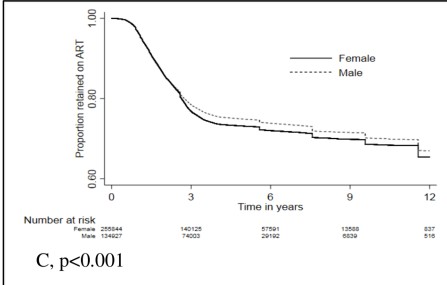

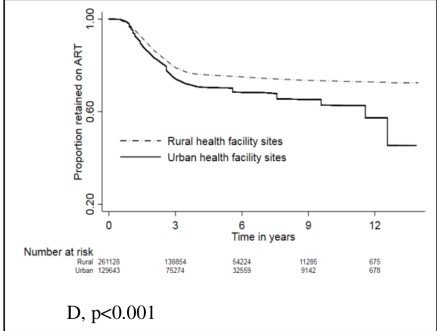

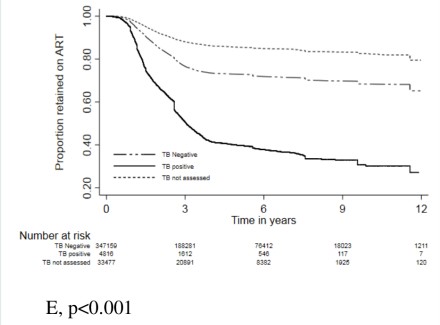

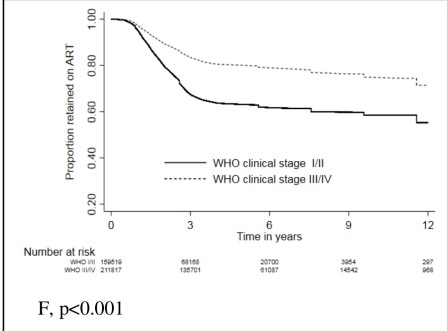

**Figure 4** The cumulative incidence probability of mortality and loss to follow-up (A), and Kaplan-Meier curves for loss to follow-up stratified by baseline age categories (B), gender (C), level of urbanisation of the health facilities (D), baseline tuberculosis infection status (E) and baseline WHO clinical stages for ART patients from the Zimbabwe national ART programme electronic patient monitoring data, 2004–2017. ART, antiretroviral therapy.

**Table 3** Multiple regression models results of LTFU among ART patients in Zimbabwe national ART programme, 2004–2017, for different models

| Characteristics | Category | Cox proportional hazard | | Parametric Gompertz model | | | Weibull model with random effects | Competing risk model of mortality on LTFU |
| | | CC+MLE (model 1) | MI+MLE (model 2) | CC+BE (model 3) | CC+MLE (model 4) | MI+MLE (model 5) | CC+MLE (model 6) | CC+MLE (model 7) |
|---|---|---|---|---|---|---|---|---|
| Health facility level | Primary healthcare | 1 (reference) | 1 (reference) | 1 (reference) | 1 (reference) | 1 (reference) | 1 (reference) | 1 (reference) |
| | Secondary hospitals* | 1.48 (1.46–1.50) | 1.51 (1.49–1.53) | 1.53 (1.51–1.54) | 1.53 (1.51–1.55) | 1.57 (1.54–1.59) | 1.48 (1.45–1.51) | 1.47 (1.45–1.50) |
| | Tertiary hospitals† | 1.17 (1.14–1.21) | 1.23 (1.20–1.27) | 1.23 (1.21–1.25) | 1.23 (1.20–1.27) | 1.30 (1.26–1.34) | 1.10 (1.06–1.14) | 1.22 (1.18–1.26) |
| Age (years) at ART initiation | 15–24 | 1.16 (1.13–1.19) | 1.17 (1.14–1.19) | 1.12 (1.11–1.14) | 1.13 (1.10–1.15) | 1.14 (1.12–1.17) | 1.23 (1.20–1.27) | 1.19 (1.16–1.21) |
| | 25–34 | 1.18 (1.17–1.20) | 1.19 (1.17–1.21) | 1.18 (1.16–1.19) | 1.18 (1.16–1.20) | 1.19 (1.17–1.20) | 1.24 (1.21–1.27) | 1.19 (1.17–1.21) |
| | 35–44 | 1 (reference) | 1 (reference) | 1 (reference) | 1 (reference) | 1 (reference) | 1 (reference) | 1 (reference) |
| | 45+ | 0.88 (0.87–0.89) | 0.87 (0.86–0.89) | 0.88 (0.87–0.89) | 0.88 (0.86–0.90) | 0.87 (0.85–0.88) | 0.86 (0.84–0.88) | 0.87 (0.85–0.89) |
| Degree of urbanisation | Rural | 1 (reference) | 1 (reference) | 1 (reference) | 1 (reference) | 1 (reference) | 1 (reference) | 1 (reference) |
| | Urban | 1.70 (1.68–1.72) | 1.67 (1.65–1.27) | 1.67 (1.65–1.69) | 1.67 (1.65–1.70) | 1.64 (1.61–1.66) | 2.06 (2.01–2.10) | 1.61 (1.59–1.63) |
| Calendar time of starting ART | 2004–2007 | 0.04 (0.04–0.05) | 0.04 (0.04–0.05) | 0.02 (0.02–2.02) | 0.02 (0.01–0.02) | 0.02 (0.01–0.02) | 0.01 (0.01–0.01) | 0.09 (0.09–0.10) |
| | 2008–2011 | 0.07 (0.7–0.08) | 0.07 (0.07–0.08) | 0.04 (0.04–0.04) | 0.04 (0.0–0.04) | 0.05 (0.04–0.05) | 0.02 (0.02–0.02) | 0.11 (0.10–0.11) |
| | 2012–2014 | 1 (reference) | 1 (reference) | 1 (reference) | 1 (reference) | 1 (reference) | 1 (reference) | 1 (reference) |
| | 2015–2017 | 8.48 (8.34–8.63) | 8.40 (8.26–8.54) | 5.95 (5.90–6.00) | 5.97 (5.88–6.06) | 5.90 (5.81–5.99) | 16.09 (15.56–16.63) | 6.02 (5.91–6.12) |
| WHO staging at ART initiation | I/II | 1 (reference) | 1 (reference) | 1 (reference) | 1 (reference) | 1 (reference) | 1 (reference) | 1 (reference) |
| | III/IV | 0.73 (0.72–0.74) | 0.74 (0.73–0.75) | 0.73 (0.73–0.74) | 0.73 (0.72–0.75) | 0.74 (0.73–0.75) | 0.67 (0.65–0.68) | 0.74 (0.73–0.75) |
| Treatment regimen at the time of data extraction | D4T(30)+3TC+NVP | 5.96 (5.72–6.21) | 6.00 (5.77–6.25) | 6.06 (5.92–6.22) | 6.09 (5.84–6.34) | 6.06 (5.82–6.30) | 15.05 (13.98–16.21) | 5.47 (5.52–5.75) |
| | D4T(30)+3TC+EFV | 8.55 (7.21–10.12) | 8.34 (7.09–9.90) | 8.71 (8.21–9.20) | 8.47 (7.16–10.03) | 8.29 (7.01–9.79) | 19.88 (15.23–25.87) | 7.98 (6.41–9.94) |
| | AZT+3TC+NVP | 0.73 (0.67–0.80) | 0.72 (0.66–0.79) | 0.74 (0.71–0.78) | 0.74 (0.67–0.81) | 0.72 (0.65–0.79) | 0.74 (0.66–0.83) | 0.75 (0.68–0.83) |
| | AZT+3TC+EFV | 1.19 (1.02–1.40) | 1.19 (1.01–1.39) | 1.19 (1.16–1.23) | 1.21 (1.03–1.42) | 1.21 (1.03–1.40) | 1.27 (1.03–1.56) | 1.09 (0.91–1.30) |
| | TDF+3TC+NVP | 2.41 (2.36–2.47) | 2.46 (2.40–2.51) | 2.46 (2.42–2.50) | 2.45 (2.40–2.51) | 2.50 (2.44–2.55) | 3.58 (3.45–3.70) | 2.29 (2.23–2.34) |
| | TDF+3TC+EFV | 1 (reference) | 1 (reference) | 1 (reference) | 1 (reference) | 1 (reference) | 1 (reference) | 1 (reference) |
| | Other first lines | 1.97 (1.90–2.04) | 1.97 (1.91–2.04) | 1.88 (1.83–1.93) | 1.89 (1.82–1.95) | 1.89 (1.83–1.96) | 2.44 (2.32–2.57) | 1.61 (1.54–1.68) |
| | Second line | 0.46 (0.42–0.49) | 0.45 (0.42–0.49) | 0.45 (0.43–0.47) | 0.44 (0.41–0.48) | 0.44 (0.41–0.47) | 0.39 (0.36–0.42) | 0.48 (0.44–0.52) |
| Tuberculosis status at ART initiation | Negative | 1 (reference) | 1 (reference) | 1 (reference) | 1 (reference) | 1 (reference) | 1 (reference) | 1 (reference) |
| | Positive | 2.14 (2.05–2.24) | 2.13 (2.04–2.23) | 2.03 (1.99–2.09) | 2.02 (1.94–2.11) | 2.01 (1.93–2.10) | 2.66 (2.50–2.83) | 1.53 (1.45–1.62) |
| | Not assessed | 0.58 (0.56–0.59) | 0.59 (0.57–0.61) | 0.60 (0.59–0.60) | 0.60 (0.58–0.62) | 0.61 (0.59–0.63) | 0.51 (0.49–0.53) | 0.60 (0.58–0.62) |

Continued

**Table 3** Continued

| Characteristics | Category | Cox proportional hazard | | Parametric Gompertz model | | | Weibull model with random effects | Competing risk model of mortality on LTFU |
|---|---|---|---|---|---|---|---|---|
| | | CC+MLE (model 1) | MI+MLE (model 2) | CC+BE (model 3) | CC+MLE (model 4) | MI+MLE (model 5) | CC+MLE (model 6) | CC+MLE (model 7) |
| Body functional status at ART initiation | Working | 1 (reference) | 1 (reference) | 1 (reference) | 1 (reference) | 1 (reference) | 1 (reference) | 1 (reference) |
| | Ambulatory | 1.22 (1.19–1.24) | 1.23 (1.20–1.26) | 1.23 (1.21–1.27) | 1.23 (1.20–1.27) | 1.25 (1.22–1.28) | 1.28 (1.23–1.32) | 1.08 (1.05–1.11) |
| | Bed ridden | 1.34 (1.17–1.53) | 1.33 (1.16–1.52) | 1.32 (1.29–1.35) | 1.34 (1.16–1.53) | 1.32 (1.15–1.51) | 1.42 (1.18–1.72) | 0.64 (0.54–0.75) |
| | AIC | 196 416.4 | | | 379 948.9 | | 445 559.7 | **195 708.7** |
| | BIC | 196 439.1 | | | 380 197.6 | | 445 819.3 | **195 731.4** |
| | DIC | | | 379 925.9 | | | | |

Bold face=low values of the information criterion.
*District and mission hospitals.
†Provincial and referral hospitals.
AIC, Akaike's information criterion; ART, antiretroviral therapy; AZT, zidovudine; BE, Bayesian estimation; BIC, Bayesian information criterion; CC, complete case; DIC, deviance information criterion; D4T, stavudine; EFV, efavirenz; MI, multiple imputation; MLE, maximum likelihood estimation; NVP, nevirapine; 3TC, lamivudine; TDF, tenofovir.

social deprivations faced by adolescents through poverty reduction and economic development, improved access to healthcare, access to health services, reduced stigma and discrimination and improved caregiver psychosocial and physical well-being. Another recommendation is the use of health facility-based peer-supporters programmes for adolescents to improve ART adherence and, ultimately, viral suppression.[43 44] The peer support activities involve peer-to-peer counselling, peer support groups and treatment buddy programmes among adolescents with the aim of ART adherence and ultimately achieving viral suppression, reducing the HIV-related illnesses among adolescents. As adolescents may exit the HIV care through becoming LTFU or withdrawals due to transition adjustments challenges from paediatric care to adult care, an early multidisciplinary and developmentally suitable transition preparedness mechanism is an alternative. The mechanism should be a comprehensive package addressing inherent adolescents' health issues like cognitive health, ART adherence, stigma, disclosure and socio-economic issues.[38 44] Besides, a simple regimen, directly observed therapy and technological interventions like cellphone reminders have been found to increase ART adherence in adolescents.

We observed an overwhelming association of tuberculosis infection and becoming LTFU that concurs with an earlier study that reported that the risk of becoming LTFU was two times more likely in HIV patients with tuberculosis infection compared with those who did not have tuberculosis.[29] This finding can be explained by overlapping toxicity of HIV and tuberculosis drugs, leading to adverse reactions that have a direct consequence of becoming LTFU.[29] Therefore, a push towards integrated care of HIV and tuberculosis service delivery would elevate HIV uptake and tuberculosis screening and reduce travel inconveniences on patients.[34 45] Also, tuberculosis treatment before ART initiation reduces the risk to LTFU[42]; hence, it is profound to increase efforts in tuberculosis screening and treatment of patients with HIV. However, issues around tuberculosis drug resistance cannot be ignored. Nowadays, HIV studies also consider information on non-communicable diseases like cancer, diabetes and hypertension; however, this information was missing from this current study database.

We also detected that patients classified in WHO stage III or IV were less likely to become LTFU relative to those classified in WHO stage I or II. However, our finding contradicts with earlier studies that have reported that experiencing a severe AIDS-defining ailment was associated with reduced risk of becoming LTFU.[27 46] Our study finding may put forward the fact that there could have been an upsurge of health-seeking behaviour among patients with advanced WHO clinical stages in this database cohort or this finding could be attributed to expansions of community awareness through health promotion in these health facilities communities.

## Strengths and weaknesses of this study

This study highlights several strengths. A regularisation variable selection technique was used to identify important covariates. The lasso model shrinks the variable coefficient to zero and produces a simpler model that can be interpreted easily. For the survival analysis, the standard Cox regression model overestimated the risk in the presence of CRs, and we used the Fine and Gray CR model instead.[47] Ignoring CRs alters the probabilities of occurrence of the outcome giving biased estimates. Standard survival methods normally used in many studies censor the CR; however, this is improper because those who have died can no longer become LTFU. Such analysis will either overestimate or underestimate the risk ratios. We also used individual data that enabled exploration and adjustment for patient-level covariates associated with the outcome to get robust information on the ART outcome patterns as compared with aggregated data.[33 48] The study follow-up period was relatively long (13 years), and the data came from all the 10 provinces in Zimbabwe without restriction to specific geographical areas. The data represented quaternary, tertiary, secondary and primary health facilities from private and public sectors. This makes our results generalisable to the whole of Zimbabwe since the data reflect a true representation of all PLWHIV in the country. To the best of our knowledge, this study has used one of the largest datasets to analyse adult ART outcomes at the patient level.

Our study results should be inferred in light of some limitations. Missing data was of concern, particularly to those variables that influence LTFU like viral load and CD4 cell count in addition to non-communicable diseases comorbidities. The presence of missing data could be explained by poor documentation of information[34] or limited access to the point of care machines for viral load and CD4 cell counts measurements.[4 49 50] The missing patterns for some covariates like viral load and CD4 cell count measurements were likely to be informative and missing not at random since differentiated care monitoring approach was used in some years covered by this study.[51] Therefore, to maintain sufficient power, covariates with a high percentage of missing observations were excluded at the multivariable analysis stage. In future, we recommend that the data clerks should be trained on the importance of capturing full information and to the programme managers to consider expanding the HIV routine data collection tool to capture non-communication diseases conditions of the HIV patients. The sociodemographic and clinical characteristics in the database were only taken at baseline or last observed at the time of data extraction; hence, we could not incorporate the time dependency of some covariates. We also acknowledge that there could be differential misclassification of the LTFU outcome in this study in the sense that some of those recorded as LTFU may be 'silent transfers' who have enrolled in a different health facility as a result of ART decentralisation and have been given a different unique identifier number; hence, this might have biased our LTFU estimates in this study. There is also a possibility of participant selection bias since we only considered patients whose information as captured through the ePMS; as a result, the reported findings could be systematically different from those patients not included. However, it is of paramount importance to note that the dataset used was huge, and representation of the health facilities in Zimbabwe was reflected; hence, our findings could be generalised to all ART services health facilities in Zimbabwe.

While our findings for the common predictors of LTFU may be comparable with earlier studies conducted in sub-Saharan African region, we had also hypothesised that those ART patients who enrolled around 2008 would have a higher risk of becoming LTFU. This was done taking note of the hyperinflation that occurred in Zimbabwe during the part of the study period under review. The economic turmoil was allied to the increased mass departure of qualified health professionals and scarcities of vital medical supplies in health facilities.[21 22] Despite this situation, we detected a protective effect of becoming LTFU between 2008 and 2011 in this study. This could be explained by the support the country received through Global Fund,[4] and ART sites increased from 282 sites in 2008 to 1006 in 2012.[37] For further studies, we recommend the use of time-varying covariates like regimen type, CD4 cell counts, WHO stages and viral load measurements and the use of spatial analysis of ART outcome to identify high burdened regions that the policymakers may use to allocate resources, especially in resource-constrained settings.

## Conclusion

The results from this current study using the data spanning over 13 years agree with the evident decline of the HIV incidence and prevalence in Zimbabwe, and the findings are comparable with similar HIV studies' findings, even from Demographic Health Survey.[52] Our results indicate that patient retention in care is still a drawback in HIV prevention and control. Policymakers and programme managers should come up with strategies to retain patients on ART. These strategies may include increased patient surveillance and strengthened patient tracing mechanisms to reduce LTFU at health facilities, strengthened community-based caregiver and peer support,[14] and nutritional aid support especially for adolescents and young adults (15–34 years). In addition, interventions that minimise 'leakages' in the HIV treatment cascade including the use of ART adherence clubs to reduce ART resistance[53] and advice on sexual abstinence should remain the cornerstone of HIV prevention and control. As much as most of the observed LTFU outcome at higher levels of care (district, provincial and central hospitals) could be 'silent transfers', most patients who become LTFU are likely to have poor ART adherence if not dead. Poor ART adherence leads to poor viral suppression that affects achieving the third 90% UNAIDS targets, which state that 90% of those patients on ART

should be virally suppressed. Clinicians should provide continual educational support on the importance of ART adherence to increase life expectancy among the HIV population and other benefits of ART. More important than not meeting the UNAIDS targets is the impact that poor viral suppression has on patients' health.

**Acknowledgements** Our acknowledgements go to the Ministry of Health and Child Care, AIDS/TB Units Department for the support in data compilation and extraction for this study. We would also like to thank the Division of Epidemiology and Biostatistics at the School of Public Health for their assistance in the getting ethical approval of this study. Thanks to Dr Nomathemba Chandiwana for technical assistance of this work.

**Contributors** ZMZ conceptualised the concept, cleaned and analysed the data and wrote the initial draft of the manuscript. JT and TC advised on analysis; EM guided and oversaw the analysis and reviewed the manuscript. All authors contributed to the interpretation of the results. All the authors reviewed the final manuscript for submission.

**Funding** This work was supported by the Developing Excellence in Leadership, Training and Science (DELTAS) Africa Initiative Sub-Saharan Africa Consortium for Advanced Biostatistics (Grant No. DEL-15-005). The DELTAS Africa Initiative is an independent funding scheme of the African Academy of Sciences (AAS) Alliance for Accelerating Excellence in Science in Africa and is supported by the New Partnership for Africa's Development Planning and Coordinating Agency (NEPAD Agency) with funding from the Wellcome Trust (Grant No. 107754/Z/15/Z) and the UK government.

**Disclaimer** The views expressed in this publication are those of the authors and not necessarily those of the AAS, NEPAD Agency, Wellcome Trust, the UK government or the Ministry of Health and Child Care, Zimbabwe.

**Competing interests** None declared.

**Patient and public involvement** Patients and/or the public were not involved in the design, or conduct, or reporting, or dissemination plans of this research.

**Patient consent for publication** Not required.

**Ethics approval** Ethics approval for this study was granted by The University of Witwatersrand's Human Research Ethics Committee (Medical) (Clearance Certificate No. M170673). Data approval was obtained from the Ministry of Health and Child Care in Zimbabwe, who are the custodians of the countrywide database.

**Provenance and peer review** Not commissioned; externally peer reviewed.

**Data availability statement** Data may be obtained from a third party and are not publicly available. The data used for this study can be found from a third party through an application process to the Zimbabwe Ministry of Health and Child Care through the HIV/AIDS Unit, which oversees the data collection and compilation process for the ART program; therefore, the data are not publicly available.

**ORCID iDs**
Zvifadzo Matsena Zingoni http://orcid.org/0000-0002-7993-1187
Tobias Chirwa http://orcid.org/0000-0003-4344-0842
Jim Todd http://orcid.org/0000-0001-5918-4914
Eustasius Musenge http://orcid.org/0000-0002-3382-2372

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
