## [Reviewer comments · BMJ Open]

ARTICLE DETAILS

TITLE (PROVISIONAL)	Competing risk of mortality on loss to follow-up outcome among HIV Patients on ART: A retrospective cohort study from the Zimbabwe National ART Programme
AUTHORS	Matsena Zingoni, Zvifadzo; Chirwa, Tobias; Todd, Jim; Musenge, Eustasius

VERSION 1 – REVIEW

REVIEWER	Eleni Verykoui Aristotle University of Thessaloniki, Greece
REVIEW RETURNED	28-Dec-2019

GENERAL COMMENTS	This is a study describing the competing risk of mortality on LTFU outcome adjusting for health care facility types amongst ART patients in Zimbabwe. The article is very clear and well organized. However, the comments below should be taken into account. p.6 line 32: The whole paragraph needs to be more informative: 1) More details should be given on the type of multiple imputation used in the study (missing at random (MAR), missing completely at random (MCAT) etc). 2) What priors were used for the Bayesian estimation? Table 3 is useful and informative; complementing this with a figure illustrating the models (e.g. in terms of model calibration) would provide a straightforward, visual assessment. In the same table, AIC and BIC for model 1 are a lot larger than those derived from the other models. The authors should double-check whether these numbers are the correct. Figure 1: Second box from the end does not include the number of patients <15 years of age or the number of those with missing baseline age whereas all the previous boxes include such information. For consistency reasons the authors could provide these numbers as well.
--

REVIEWER	Claudia Cortes Universidad de Chile & Fundacion Arriaran
REVIEW RETURNED	23-Jan-2020

GENERAL COMMENTS	"Therefore, some of the reported LTFU in higher levels of care may be misclassification due to "silent transfers" or deaths among much ill patients through the referral system" this sentence from the discussion section is the main difficulty I find in your work. If you are willing to determinate the rate of LTFU you must be sure those patients you are classifying as abandonment are not dead. even it was mentioned as a weakness of the study, need to look
--

	after that data. Maybe cross check with national death registry? because if it is just supported by information given my close contacts or family it is a great bias.
--	---

REVIEWER	Ferdinand C Mukumbang South African Medical Research Council South Africa
REVIEW RETURNED	10-May-2020

GENERAL COMMENTS	Competing risk of mortality on loss to follow-up outcome among HIV Patients on ART: A retrospective cohort study from the Zimbabwe National ART Programme Thank you for the opportunity to review this article: Overall, this is a very relevant and useful study and definitely has policy implications for the continuous treatment and care of PLHIV in Zimbabwe. While the study is also scientifically well-conceived and executed, I have identified some issues that should be addressed to improve the submission. It is important that the authors mention that the Kaplan-Meier description and Cox regressions were conducted. These, in my opinion, were the central analyses conducted and should thus be reported in the abstract. Considering that the authors sought to determine the effect of health facility type after ART services decentralization on loss to follow up, it is important that the authors should adequately describe the different facility types that exist in the Zimbabwean health system to help the naïve reader. Also, the discussion section of the study hardly considered the different rates of LTFU and the implication that this has on services delivery organization. Page 6: The authors wrote: “There was a systematic selection of the health facilities which represent all levels of care in Zimbabwe to implement ePMS.” It is unclear what the authors mean by this statement and especially what is meant by “systematic selection”. More details should be provided to indicate what is actually meant by this. Figure 3. The LTFU seem to peak in the month of January. Is there is possible examination for this observation and are there any public health implications for this? One would imagine that at a certain point, differentiated care models were implemented in Zimbabwe to improve the retention in care of people living with HIV. Considering that the rate of LTFU increased with time, can one argue with for impact of these models based on the study findings?
---

REVIEWER	Victoria Boggiano UNC Family Medicine Chapel Hill, United States
REVIEW RETURNED	14-May-2020

GENERAL COMMENTS	Overall, very important topic as loss to follow up remains a major issue standing in the way of the global fight against HIV/AIDS. There is some very interesting data in this paper, but I think there should be further clarification of the study question and the importance of the findings.
---

	Introduction:  - Very good summary of the importance of understanding ART use and viral suppression globally and in Zimbabwe! - Consider having native english speaker read through and edit to avoid grammatical errors and for clarity; for ex, the second paragraph has a clause that reads "...however, the last 90% estimated at 73% in 2018." It could be revised to say "however, limitations exist. For instance, only 73% of individuals living with HIV in Zimbabwe who are on ART have achieved viral suppression." - When were ART services in Zimbabwe decentralised? - I am confused about what exactly your study is aiming to prove. That the type of health facility a patient receives care at affects loss to follow up and therefore mortality? That there are certain patient characteristics that affect LTFU? More clarity is needed in the introduction. Methods:  - Figure 1: What does "ART initiation date of death after 2017" mean? - Please provide more information about the electronic patient management system in Zimbabwe. Is - Please clarify in the paragraph under "Study design and data sources" exactly what the first line regimen is, as the way it is currently written is unclear. Is it a combination of two NRTI and one non-NRTI, so three drugs in total? - How did you choose the predictors considered in the final regression model? - Can you say more about how you incorporated time into your adjustment, and which economic turmoil you are referring to (or include a citation)? - Ethical approval section: Please specify whether the Ministry of Health and Child Care is from Zimbabwe. - In the introduction or methods section please spend some time outlining the way the health system is set up as it ties into the results section in terms of reporting LTFU. Results:  - Please specify when using phrases like "in WHO stage III/IV" - Why was baseline CD4 cell count only available in 35.3% of the patients? - As above, please define in intro or methods the differences between primary health care, district/mission hospitals, and provincial/referral hospitals and how patients were allocated among each. Any concern for overlap of patients between the different categories? Discussion:  - Overall hard to understand exactly what the main points are in the beginning of the discussion, at least partially due to language barrier - recommend working on clarifying in the second paragraph exactly what the findings were and what they represent. For example, the phrase "poor unaccountability of death outcome" is difficult to understand. - Please provide again more clarification of what the various levels of care are and how this affects follow-up. - Do really like the points made about association of LTFU and age, and LTFU and tuberculosis history
--	---

REVIEWERS	Authors' response	Page
Reviewer 1: Eleni Verykoui Institution and Country: Aristotle University of Thessaloniki, Greece		
Comment 1: This is a study describing the competing risk of mortality on LTFU outcome adjusting for health care facility types amongst ART patients in Zimbabwe. The article is very clear and well organized. However, the comments below should be taken into account.	Thank you very much for the comment.	
Comment 2: p.6 line 32: The whole paragraph needs to be more informative: 1) More details should be given on the type of multiple imputation used in the study (missing at random (MAR), missing completely at random (MCAT) etc). 2) What priors were used for the Bayesian estimation?	Thank you very much for the comment. We have provided additional details to this paragraph. We used normal priors, and this has been added in the manuscript.	Pg 8 line 226-227 Pg 8 line 230-232
Comment 3: Table 3 is useful and informative; complementing this with a figure illustrating the models (e.g. in terms of model calibration) would provide a straightforward, visual assessment.	Thank you very much for your comment. A figure has been inserted in the appendix section to support Table 3.	Pg 12 Line 286-288
Comment 4: In the same table, AIC and BIC for model 1 are a lot larger than those derived from the other models. The authors should double-check whether these numbers are the correct.	Thank you for the comment. We have double-checked the values.	Pg 13 Table 3
Comment 5: Figure 1: Second box from the end does not include the number of patients <15 years of age or the number of those with missing baseline age whereas all the previous boxes include such information. For consistency reasons the authors could provide these numbers as well.	Thank you very much for the comment. We have included that information in Figure 1.	Figure 1
Reviewer 2: Claudia Cortes Institution and Country: Universidad de Chile & Fundacion Arriaran		
Comment 1: "Therefore, some of the reported LTFU in higher levels of care may be misclassification due to "silent transfers" or deaths among much ill patients through the referral system". This sentence from the discussion section is the main difficulty I find in your work. If you are willing to determinate the rate of LTFU you must be sure those patients you are classifying as	Thank you very much for the comment. What we are saying is that some of those recorded as LTFU may be "silent transfers" who have enrolled in a different clinic and been given a different unique identifier. We also to determine if there were possible duplicate using a collection of some variables and not necessarily the unique identifier to clean the data	Pg 19 line 466-470

abandonment are not dead. Even it was mentioned as a weakness of the study, need to look after that data. Maybe cross check with national death registry? Because if it is just supported by information given my close contacts or family it is a great bias.	further. Moreover, there is a possibility that some of these LTFU patients might have died at the time of our analysis. But this paper is looking at death as a competing risk for LTFU. Therefore, the question as to whether LTFU leads to death is not as easy. The assumption is that LTFU is censoring of the patient time, and they would still be at risk of death following LTFU. With the information we have, we cannot determine whether the mortality rate in those LTFU is the same as the mortality rate in those under observation.	
Reviewer 3: Ferdinand C Mukumbang South African Medical Research Council, South Africa		
Comment 1: Thank you for the opportunity to review this article: Overall, this is a very relevant and useful study and definitely has policy implications for the continuous treatment and care of PLHIV in Zimbabwe. While the study is also scientifically well-conceived and executed, I have identified some issues that should be addressed to improve the submission.	Thank you very much.	
Comment 2: It is important that the authors mention that the Kaplan-Meier description and Cox regressions were conducted. These, in my opinion, were the central analyses conducted and should thus be reported in the abstract.	Thank you for the comment. We have included these in the abstract methods section.	Pg 1 line 26
Comment 3: Considering that the authors sought to determine the effect of health facility type after ART services decentralization on loss to follow up, it is important that the authors should adequately describe the different facility types that exist in the Zimbabwean health system to help the naïve reader.	Thank you very much for the comment. We have included this information in the manuscript.	Pg 3 and 4 line 86-97
Comment 4: Also, the discussion section of the study hardly considered the different rates of LTFU and the implication that this has on services delivery organization.	Thank you very much for the comment. We have commented on this in the discussion.	Pg 15 and 16 line 330-365
Comment 5: Page 6: The authors wrote: "There was a systematic selection of the health facilities which represent all levels of care in Zimbabwe to implement ePMS." It is unclear what the authors mean by this statement and especially what is meant by "systematic selection". More details should be provided to indicate what is actually meant by this.	Thank you very much for the comment. We have rephrased this sentence in the manuscript.	Pg 6 line 157-159
Comment 6:		

Figure 3. The LTFU seem to peak in the month of January. Is there is possible examination for this observation and are there any public health implications for this? One would imagine that at a certain point, differentiated care models were implemented in Zimbabwe to improve the retention in care of people living with HIV. Considering that the rate of LTFU increased with time, can one argue with for impact of these models based on the study findings?	Thank you very much for the comment. Maybe there could be possible public health implications of these high peaks in certain months. One possibility could be the issue of high mobility of people during the festive period, which may lead to relocation contributing to the LTFU outcome. Such processes might not be captured through the patient tracking system if the patient is assigned a different unique identifier. Truly, the differential models have done so well in terms of ART coverage in Zimbabwe and other sub-Saharan African countries. In terms of ART retention, there is a decrease in the attrition rates in general across most HIV studies, and this was evident in our study which had a lower LTFU rate compared to other countries with a similar setting as Zimbabwe. Absolutely, the differential models showed a positive impact on patient retention in care of PLWHIV. The increase in LTFU rates over time, peaking in the deferential models period could be explained by the "silent transfers" as pointed out in the manuscript since most of the LTFU occurred in higher levels of care facilities. There could also be a possibility that the data were increasing over time as more health facilities were introduced to ePMS with an improved tracking system and better reporting compared to the period whereby paper-based system was solely used. Also, the peaks in some time points could be an artefact of the LTFU observations. That is, those LTFU in the early years have more time for their status to be confirmed; hence, low peaks of LTFU rates as some may die and have their status updated as death or they may return to the clinic and be identified as no longer LTFU while the high peak in most recent years (2017) could be people who may have missed their appointment and classified as LTFU but may return.	Pg 4 line 100-102 Pg 14 line 322-323 Pg 15 line 341-344
Reviewer 4: Victoria Boggiano Institution and Country: UNC Family Medicine Chapel Hill, United States		
Overall, very important topic as loss to follow up remains a major issue standing in the way of the global fight against HIV/AIDS. There is some very interesting data in this paper, but I think there should	Thank you very much for the comment.	

be further clarification of the study question and the importance of the findings.		
Comment 1: Introduction: - Very good summary of the importance of understanding ART use and viral suppression globally and in Zimbabwe! - Consider having native English speaker read through and edit to avoid grammatical errors and for clarity; for ex, the second paragraph has a clause that reads "...however, the last 90% estimated at 73% in 2018." It could be revised to say "however, limitations exist. For instance, only 73% of individuals living with HIV in Zimbabwe who are on ART have achieved viral suppression." - When were ART services in Zimbabwe decentralised? - I am confused about what exactly your study is aiming to prove. That the type of health facility a patient receives care at affects loss to follow up and therefore mortality? That there are certain patient characteristics that affect LTFU? More clarity is needed in the introduction.	Thank you very much for the comments. This has been changed as suggested. In 2010/11 and this has been added in the manuscript. We aim to determine the LTFU pattern in this cohort after the decentralization of ART services from higher levels of care to lower levels of care. This means we assess LTFU at different health facility levels so that we see which health facilities tend to experience high rates of LTFU. In addition to this, we also adjusted for other individual characteristics. We have reworked the introduction section accordingly.	Pg 3 line 82-83 Pg 4 line 100 Pg 4 line 107-114
Comment 2: Methods: - Figure 1: What does "ART initiation date of death after 2017" mean? - Please provide more information about the electronic patient management system in Zimbabwe is? - Please clarify in the paragraph under "Study design and data sources" exactly what the first line regimen is, as the way it is currently written is unclear. Is it a combination of two NRTI and one non-NRTI, so three drugs in total - How did you choose the predictors considered in the final regression model? - Can you say more about how you incorporated time into your adjustment, and which economic turmoil you are	Thank you for the comments. We have corrected that error. More information has been added on ePMS as suggested. We have noted the comment and have added that information in the manuscript. Yes, it is a three drug combination therapy. We used variable selection techniques stated in the data analysis section and supported by Figure 2. We use the calendar time covariate, which we adjusted for in the model in addition to time since ART	Figure 1 Pg 5 and 6 Line 140-175 Pg 5 line 132-133 Pg 7 line 197-200 Pg 8 line 219-221

referring to (or include a citation)? - Ethical approval section: Please specify whether the Ministry of Health and Child Care is from Zimbabwe. - In the introduction or methods section please spend some time outlining the way the health system is set up as it ties into the results section in terms of reporting LTFU.	initiation captured in the model specification. The economic turmoil we are referring to is when the country experience mass departure of medical personnel and scarcity of medical supplies, including HIV supplies due to economic challenges at that time. We have now stated that. Noted, we have added that information in the introduction section.	Pg 9 line 247 Pg 3 and 4 line 86-97
Comment 3: Results: - Please specify when using phrases like "in WHO stage III/IV" - Why was baseline CD4 cell count only available in 35.3% of the patients? - As above, please define in intro or methods the differences between primary health care, district/mission hospitals, and provincial/referral hospitals and how patients were allocated among each. Any concern for overlap of patients between the different categories?	Thank you for the comments. We have explained that in the manuscript The baseline CD4 cell counts were not recorded for every participant we included in this study; hence, only a few have a CD4 cell count measurement. Also, the study covers a time when ART initiation was based on clinical classification, and then differential monitoring later introduced for these laboratory assessments. This could explain the missing baseline CD4 cell count measurements in some participants. We have added the information in both the introduction and the methods sections for clarity. We have added the information in both the introduction and the methods sections for clarity	Pg 11 Line 266 Pg 3 and 4 line 86-97 Pg 7 and 8 line 203-209
Comment 4: Discussion: - Overall hard to understand exactly what the main points are in the beginning of the discussion, at least partially due to language barrier - recommend working on clarifying in the second paragraph exactly what the findings were and what they represent. For example, the phrase "poor unaccountability of death outcome" is difficult to understand. - Please provide again more clarification of what the various levels of care are and how this affects follow-up.	Thank you very much for the comments. We have revised the discussion section accordingly. We have removed that statement in this section. More detail provided in the introduction and methods sections of what the levels of health care are.	Pg 14 and 15 line 319-329 Pg 3 and 4 line 86-97 Pg 15 and 16

- Do really like the points made about association of LFTU and age, and LFTU and tuberculosis history.	Health facility effect of LTFU has been discussed Thank you for the comment.	line 330-365
---	---	---------------------

VERSION 2 – REVIEW

REVIEWER	ELENI VERYKOUKI Aristotle University of Thessaloniki
REVIEW RETURNED	01-Jul-2020

GENERAL COMMENTS	The authors have revised the manuscript adequately. I have no further comments.
--

REVIEWER	Victoria Boggiano MD MPH University of North Carolina at Chapel Hill
REVIEW RETURNED	30-Jun-2020

GENERAL COMMENTS	This draft is improved compared to prior and the authors are providing important insight into a topic that remains vitally important as we seek to battle the global HIV/AIDS epidemic. There remain some gaps that I think need to be addressed, but this draft is much closer to a final version. I recommend that a native English speaker read through the manuscript and edit for clarity. Specific comments by section below: Introduction:  - What do you mean at the start of the second paragraph, with the phrase “rigorous analyses are restricted”? Who is restricting them - the government? Please clarify. - Very good definitions of the different levels of care in Zimbabwe. - Please state clearly which level ART services were decentralized to — primary vs secondary care categories or both? Who oversaw this decentralization? - Why is loss to follow up more likely to persist after decentralization, as you state in the second to last paragraph of the introduction? - Please explain your rationale for your study period as it does include the decentralization in ART services that occurred in 2010-2011. Methods:  - What was the HIV prevalence at the start of your study period in 2004? - How many ART sites were there at the start in 2004? - The monitoring and evaluation section should be condensed and edited for clarity. And just to clarify, the ePMS system is just for ART? How does it interact with other health care data systems in Zimbabwe? And please also specify whether ART-related patient data data prior to 2012 was also included in the ePMS — it appears that it was, but would be good to state specifically. - Why did you group quarternary (referral) and tertiary (provincial) levels of care in your analysis? - It seems like the data you had from 2012-2017 was much more robust because of the ePMS system. Is this the case? If so how did you account for that in your analysis?
--

	- Please explain in more detail the predictors you adjusted for. For instance, what is body functional status? Results:  - Paragraph 1 —> when you say the patients “transferred out,” what are you referring to? Where did they transfer to? - In addition to gender and age, I would specify average body functional status which is more meaningful than a patient’s weight. - Within the functional status category, what does “missing” mean? This information was not available? Please clarify. - Clarify what you mean by WHO stages. What does WHO stage III or IV mean? Discussion:  - Please clarify what you mean by the phrase “silent transfers.” Is it referring to the instance where a patient changes where they are getting their care but the system improperly picks it up or does not track it? - How has the ePMS worked to try to improve its tracking system? How did patient tracking evolve over the course of your study period and did that have any impact on your results? - One important point that needs to be clarified: If a patient is more ill, and he or she is more likely to pass away from HIV, does that mean that he or she is more likely to fall into the LTFU category? Do your results show this? I would factor this into your point about sicker patients going to higher levels of care. - Does the ePMS track children as well as adults? How does tracking differ among pediatric and adult hospital systems? Conclusion:  - Please reword the first sentence for clarity. - Please explain what you mean by the phrase “silent transfers” as referenced in the discussion section as well - More important than not meeting the UNAIDS targets is the impact that poor viral suppression has on patients’ health — I recommend specifying this.
--	---

VERSION 2 – AUTHOR RESPONSE

REVIEWERS	Authors' response	Page
Reviewer 4: Victoria Boggiano Institution and Country: UNC Family Medicine Chapel Hill, United States		
This draft is improved compared to prior and the authors are providing important insight into a topic that remains vitally important as we seek to battle the global HIV/AIDS epidemic. There remain some gaps that I think need to be addressed, but this draft is much closer to a final version. I recommend that a native English speaker read through the manuscript and edit for clarity.	Thank you very much for the comment. Thank you very much for your suggestion. We are fortunate to have one of our co-authors, who is a native English speaker. He has gone through this document and made necessary suggestions and edits to the	

	manuscript for clarity.	
INTRODUCTION		
Comment 1: What do you mean at the start of the second paragraph, with the phrase “rigorous analyses are restricted”? Who is restricting them - the government? Please clarify	Thank you very much for your comment. We have rephrased this in the manuscript.	Page 3 Line 77-78
Comment 2: Very good definitions of the different levels of care in Zimbabwe.	Thank you very much.	
Comment 3: Please state clearly which level ART services were decentralized to — primary vs secondary care categories or both? Who oversaw this decentralization?	Thank you very much for your comment. We have clarified these questions in the manuscript. The government of Zimbabwe through the Ministry of Health and Child Care	Page 4 Line 100-104
Comment 4: Why is loss to follow up more likely to persist after decentralization, as you state in the second to last paragraph of the introduction?	Thank you for the comment. ART decentralisation mainly improves patients’ retention, ART adherence and access to care but does not mean there will not be LTFU. Of course, the LTFU rates may be going down, but they are not prevented from occurring completely by the decentralisation strategy. LTFU is associated with social (stigmatization, social deprivation, health literacy issues) and individual factors (perception and misconceptions over ART benefits) which still hinder retention in care which may not be mitigated by decentralization only. We have added this explanation in the manuscript.	Page 4 Line 110-112
Comment 5: Please explain your rationale for your study period as it does include the decentralization in ART services that occurred in 2010-2011.	Thank you for the comment. We have inserted the explanation to this.	Page 4 Line 114-116
METHODS		
Comment 6: What was the HIV prevalence at the start of your study period in 2004?	Thank you for the comment. It was estimated at 25%, and we have added this information in the manuscript.	Page 5 Line 125
Comment 7: How many ART sites were there at the start in 2004?	Thank you for the comment. On 2004, there were 5 ART sites, and we have added this information in the manuscript.	Page 5 Line 133
Comment 8: The monitoring and evaluation section should	Thank you for the comment, we have	Page 5-6

be condensed and edited for clarity. And just to clarify, the ePMS system is just for ART? How does it interact with other health care data systems in Zimbabwe? And please also specify whether ART-related patient data prior to 2012 was also included in the ePMS — it appears that it was, but would be good to state specifically.	condensed and edited the section. It is an all HIV-related database. We have clarified this in the manuscript. The data captured in the ePMS is further linked to the Zimbabwe E-health system. We have specified this in the manuscript.	Line 147-155 Page 6 line 168 Page 6 line 173
Comment 9: Why did you group quaternary (referral) and tertiary (provincial) levels of care in your analysis?	Thank you very much for your comment. We grouped the referral and the provincial levels of care because their patient care process is comparable.	Page 8 Line 210-211
Comment 10: It seems like the data you had from 2012-2017 was much more robust because of the ePMS system. Is this the case? If so how did you account for that in your analysis?	Thank you very much for your comment. The 2012-2017 data could be considered to have been robust since the data was compiled in real-time compared to the data before the ePMS launch, which was compiled retrospectively. Since we used the time to event models and adjusted for the enrolment time variable in our regression models, that way, we managed to account for any time variations in our analysis.	
Comment 11: Please explain in more detail the predictors you adjusted for. For instance, what is body functional status?	Thank you very much for your comment. We have explained the predictors in the manuscript.	Page 7-8 Line 202-208
RESULTS Comment 14: Paragraph 1 —> when you say the patients “transferred out,” what are you referring to? Where did they transfer to?	Thank you very much for your comment. We have clarified this in the manuscript.	Page 9 Line 260-261
Comment 15: In addition to gender and age, I would specify average body functional status which is more meaningful than a patient’s weight.	Thank you very much for the comment. We have specified the body functional status as suggested.	Page 9 line 270-271
Comment 16: Within the functional status category, what does “missing” mean? This information was not available? Please clarify.	Thank you very much for the comment. Yes, the information was not available. We have added a footnote clarifying this under Table 1 and 2.	Page 11 line 274 Page 12 line 286
Comment 17: Clarify what you mean by WHO stages. What does WHO stage III or IV mean?	Thank you very much for the comment. This is the Roman numerals	

	numbering of WHO stages whereby III=3 and IV=4.	
DISCUSSION		
Comment 18 Please clarify what you mean by the phrase "silent transfers." Is it referring to the instance where a patient changes where they are getting their care but the system improperly picks it up or does not track it?	Thank you very much for your comment. It means the patients transfer, but the system does not pick it. We have clarified this in the manuscript.	Page 15 line 348-350
Comment 19: How has the ePMS worked to try to improve its tracking system? How did patient tracking evolve over the course of your study period and did that have any impact on your results?	Thank you very much for your comment. The ePMS was designed to automatically flag for those patients who would have missed their appointment visit. After that follow-up mechanism, including the use of SMS reminders, phone calls, and home visits by community health workers should be done. However, such follow-ups still require to be strengthened to improved efficiency in the tracking system.	
Comment 20: One important point that needs to be clarified: If a patient is more ill, and he or she is more likely to pass away from HIV, does that mean that he or she is more likely to fall into the LTFU category? Do your results show this? I would factor this into your point about sicker patients going to higher levels of care.	Thank you very much for your comment. Yes, it means that they are likely to become LTFU. No, our data could not support this further analysis. This point is already factored in the manuscript.	Page 16 line 366 Page 16 Line 365-367
Comment 21: Does the ePMS track children as well as adults?	Thank you very much for your comment. Sure it tracks all age groups.	
Comment 22: How does tracking differ among pediatric and adult hospital systems?	Thank you very much for your comment. The tracking system mechanism is uniform for all HIV patients. The main goal is to make sure that the patients are consistently taking their treatment and showing up for their scheduled treatment refill visits. This is then used as a proxy to measure ART adherence and patients' retention.	
CONCLUSION		
Comment 23 Please reword the first sentence for clarity	Thank you very much for the comment. The sentence has been rephrased as suggested.	Page 20 line 499-501

Comment 24: Please explain what you mean by the phrase “silent transfers” as referenced in the discussion section as well	Thank you very much for the comment. This has been defined in the discussion section.	Page 15 line 348-350
Comment 25: More important than not meeting the UNAIDS targets is the impact that poor viral suppression has on patients’ health — I recommend specifying this.	Thank you very much for the comment. We have added this point as suggested in the manuscript.	Page 21 line 515-517